# How Fear of COVID-19 Can Affect Treatment Choices for Anaplastic Large Cell Lymphomas ALK+ Therapy: A Case Report

**DOI:** 10.3390/healthcare9020135

**Published:** 2021-01-31

**Authors:** Antonello Sica, Caterina Sagnelli, Beniamino Casale, Gino Svanera, Massimiliano Creta, Armando Calogero, Renato Franco, Evangelista Sagnelli, Andrea Ronchi

**Affiliations:** 1Department of Precision Medicine, University of Campania “Luigi Vanvitelli”, 80131 Naples, Italy; 2Department of Mental Health and Public Medicine, University of Campania “Luigi Vanvitelli”, 80138 Naples, Italy; caterina.sagnelli@unicampania.it (C.S.); evangelista.sagnelli@unicampania.it (E.S.); 3Pain Department, AORN Dei Colli—V. Monaldi, 80131 Naples, Italy; benny.casale@hotmail.com; 4Department of Medical Area ASLNA2 NORTH, 80014 Giugliano, Italy; gsvanemato@inwind.it; 5Department of Neurosciences, Reproductive Sciences and Odontostomatology, University of Naples “Federico II”, 80131 Naples, Italy; massimiliano.creta@unina.it; 6Department of Advanced Biomedical Sciences, University of Naples Federico II, 80131 Naples, Italy; armando.calogero2@unina.it; 7Pathology Unit, Department of Mental and Physical Health and Preventive Medicine, University of Campania “Luigi Vanvitelli”, 80138 Naples, Italy; renato.franco@unicampania.it (R.F.); andrea.rochi@unicampania.it (A.R.)

**Keywords:** crizotinib, anaplastic large cell lymphomas ALK+, bridge therapy in NHL ALK+, ALK+ patients, anticancer therapy

## Abstract

Background: The t (2; 5) chromosomal rearrangement of the ALK gene with nucleophosmin 1 gene (NPM1), resulting in an NPM1–ALK fusion, was first demonstrated in 1994 in anaplastic large cell lymphoma, (ALCL), a T-cell lymphoma responsive to cyclophosphamide, abriblastine, vincristine and prednisone in approximately 80% of cases; refractory cases usually respond favorably to brentuximab vedotin. These treatments are regarded as a bridge to allogeneic hematopoietic stem cell transplantation (allo-SCT). Nowadays, transplant procedures and the monitoring of chemotherapy patients proceed very slowly because the SARS-CoV-2 pandemic has heavily clogged the hospitals in all countries. Results: A 40-year-old Caucasian woman was first seen at our clinical center in June 2020. She had ALCL ALK+, a history of failure to two previous therapeutic lines and was in complete remission after 12 courses of brentuximab, still pending allo-SCT after two failed donor selections. Facing a new therapeutic failure, we requested and obtained authorization from the Italian drug regulatory agency to administer 250 mg of crizotinib twice a day, a drug incomprehensibly not registered for ALCL ALK +. Conclusions: The response to crizotinib was optimal since no adverse event occurred, and CT-PET scans persisted negative; this drug has proved to be a valid bridge to allo-SCT.

## 1. Introduction

The t (2; 5) chromosomal rearrangement and the resulting nucleophosmin (NPM1)-ALK fusion was first described in anaplastic large cell lymphoma (ALCL) in 1994 and further studies have shown that ALK rearrangements are present in approximately 55% of ALCL cases and in almost all affected children. Considering that, it would be natural that ALK inhibitors were registered for this disease, but the Italian drug regulatory agency, Agenzia Italiana del Farmaco (AIFA), authorized these drugs only for treating non-small cell lung cancer (NSCLC) [1,2]. The enormous efficacy of ALK inhibitors in the treatment of NSCLC has shown that these mutations behave like real oncodrivers and, therefore, have completely replaced the chemotherapy previously practiced in these neoplasms. For ALCL, conventional chemotherapy has shown better efficacy than ALK inhibitors in all studies performed, which probably explains why the AIFA has not submitted ALCL for approval. ALCL is a T-cell lymphoma that responds to cyclophosphamide, abriblastin, vincristine, prednisone (CHOP) in about 80% of treated patients, while the anti-CD30 monoclonal antibody brentuximab-vedotin is usually beneficial in refractory cases [3,4,5,6]. These treatments are used as a bridge to hematopoietic allogeneic stem cell transplantation (allo-SCT), which is considered the definitive therapy for refractory cases.

Nowadays, due to the SARS-CoV-2 pandemic, the strategies leading to allo-SCT or chemotherapy have changed in clinical practice [7]. In fact, the clogging of health facilities caused by this pandemic has slow down the procedures to access allo-SCT and has complicated the follow up of patients, either those with transplants or undergoing chemotherapy. These slowdowns require alternative therapeutic strategies to block the progression of the disease. Waiting for a suitable donor, we treated a 40-year-old Caucasian woman with ALCL and a history of failure to a first- and second-line therapy with crizotinib, a drug that has a specific action against ALK + and a proven specific efficacy [8,9]. This drug is not registered in Italy for this type of lymphoma, but its use for our patient was authored by the AIFA. In addition to NPM1, many other fusion partners have been described: TPM3, TFG, ATIC, CLTC, TPM4, MSN, ALO17, MYH9, TRAF1, and TPM3 [10,11]. Current knowledge suggests that the presence of oncodrivers prevails over the phenotype/diagnosis. As with non-small cell lung cancer (NSCLC), a fusion partner can influence the response to different ALK-TKIs, but we still don’t know how it works in other clinical situations with ALK rearrangement [12]. Several clinical trials are currently underway: alectinib for relapsed or refractory anaplastic lymphoma kinase-positive anaplastic large cell lymphoma [13], lorlatinib for refractory ALK-positive ALCL (NCT03505554) and for refractory neuroblastoma (NCT03107988), and brigatinib for ALK-positive ALCL (NCT03719898). These studies will lead to a better understanding of how these ALK-TKIs work in ALCL or other solid tumors with ALK rearrangement.

Current knowledge indicates that crizotinib can be effective in pediatric patients with ALK-positive ALCL [14] and in infantile hemispheric glioma with ALK rearrangement [15,16]; this drug was also effective in an adult ALK-positive-patient with advanced stage ganglioneuroblastoma. Such favorable effects of the ALK inhibitor crizotinib led us to choose this drug to create a safe bridge to allo-SCT for our adult patient with ALCL ALK+ who was no more respondent to several therapies; this was an excellent choice.

## 2. Case Report

A 40-year-old Caucasian woman with anaplastic large cell lymphoma ALK+ (ALCL ALK+) in complete remission (CR) after 16 cycles of therapy with brentuximab (1.8 mg/kg) was first observed at our clinical center in June 2020, a treatment scheduled pending allo-SCT.

The diagnosis of ALCL ALK+ was made with the microscopic examination of a latero-cervical lymph node surgically removed in October 2016, also using immunohistochemistry and fluorescence in situ hybridization (FISH). Immunohistochemistry showed intense and diffuse positivity in neoplastic cells, allowing the diagnosis of anaplastic large cell lymphoma, ALK-positive. Moreover, we indagated the specific fusion partner using a Two Color, Two Fusion Translocation Probe, designed to detect the translocation between the ALK gene located at 2p23 and the NPM1 gene located at 5q35. The patient was Stage IV, according to the Ann Arbor classification (Figure 1A), and was treated with cyclophosphamide, epirubicin, vincristine, prednisone (CEOP) for six cycles. In March 2017, as soon as these cycles finished, she had an inguinal recurrence with the presence of blasts of meningeal lymphoma. From April to August 2017, she received a second-line therapy with MTX-ARAC-THIOthepa (MAT: methotrexate, cytarabine, Thiothepa) for four cycles plus six rachicentesis with an infusion of MTX, ARAC, and steroids. Reaching remission, she underwent peripheral blood stem cells (PBSC) mobilization chemotherapy, according to the idarubicin, ARA-C, filgrastim scheme plus Plerixafor. In October 2017, she practiced PBSC apheresis and, in November 2017, therapeutic consolidation with autologous stem cell transplantation (ASCT), after high-dose chemotherapy (HDT) with a thiotepa-busulfan (Thio-Bu) regimen (Table 1), with complete remission (Figure 1B). A voluminous lymphadenopathy appeared in the right armpit at the end of May 2019, and, greatly reduced with paracetamol, we interpreted it as an early sign of progression, oddly improved with NSAIDs. In July 2019, the patient developed mild systemic low-grade irregular fever and slight weight loss; she performed all blood chemistry tests that showed the following: erythrocyte sedimentation rate (ESR) 20; Ferritin 58; β2-microglobulin 1.32; lacto dehydrogenase (LDH) 366; Hemoglobin 12.5 gr/dL; platelets 198,000 × 10^3^/mmc; white blood cells 4800 × 10^3^/mmc; 49%; leukocyte 37%; monocyte 9%. As the axillary lymphadenopathy persisted, a biopsy was performed, and the histological examination showed a new relapse of the disease (Figure 1C).

Consequently, in July 2019, she was treated with brentuximab, performed for 16 cycles until June 2020 (Figure 1D). Before starting this therapy, it was necessary to add tramadol 150 mg per day orally as analgesic therapy, because she reported continuous pain resistant to paracetamol in the right armpit [17,18].

Since CR was achieved after six months of this third line of treatment, the donor selection for allo-SCT began. A haploidentical male cousin was summoned in December 2019 to undergo the first level of investigations to be admitted for donation: HCV-Ab, HBsAg, HBcAb, HIV-Ab, blood count, creatinine, Blood urea nitrogen (BUN), aminotransferases (ATs), Aspartate transaminase (AST), amylase, lipase, bilirubin, electrolytes, triglyceride, cholesterol and glycemy, serum protein electrophoresis, LDH [19,20,21,22,23,24].

Unfortunately, he was found to be Virus Epstein-Barr (EBV) IgG positive, with anti-HLA (HLA: Human Leucocyte Antigens), and donor mismatched on loci A and B (6/8), and therefore, not completely suitable, according to the criteria of the Italian Bone Marrow Donor Registry. During the SARS-CoV-2 pandemic, the search for donors has slowed down a lot, mostly due to the fear of donors and patients accessing hospitals. In this period, aplasticizing therapies for transplants have been considered extendable for up to 8–12 weeks for non-aggressive or non-life-threatening diseases. These extensions have been suggested in the hope of a future strong reduction in the circulation of the virus. A lower likelihood of SARS-CoV-2 infection is certainly desirable for patients for whom strong immunosuppressive therapy is required. In subjects awaiting allogeneic transplantation, conditioning chemotherapy should be started only after the arrival of the donor cells and cryopreservation. As the waiting time for allogeneic transplantation, however indispensable for our patient, was extended, all possible courses of brentuximab were performed. Of note, the schedule for administering brentuximab to patients with stabilized or improving disease suggests a maximum of 16 cycles to be practiced in approximately one year.

The attempt to select a suitable donor has continued but with no success, due to the significant shortage of donors in this prolonged SARS-CoV-2 pandemic, when many people, including organ donors, fear contagion and avoid entering hospitals. In the meanwhile, an AIFA approval was requested to administer 250 mg of crizotinib twice daily, a drug not registered for treating ALCL ALK+ in Italy. The approval was given in February 2020 and treatment began one month after the last dose of brentuximab. The response to the new treatment was optimal since no side effects have occurred and the [18f] FDG-Positron Emission Tomography/Computed Tomography ([18f] FDG-PET-CT) performed in September 2020 was negative (Figure 1D).

A haploidentical female cousin was summoned to undergo the first level of investigations (HIV-Ab, CMV-Ab, HCV-Ab, HBsAg, HBcAb, EBV-Ab, blood count, creatinine, BUN, AST, ALT, amylase, lipase, bilirubin, electrolytes, triglyceride, cholesterol and glycemy, serum protein electrophoresis) to be admitted for donation in October 2020 [25,26,27,28]. Unfortunately, she was EBV-Ab IgM positive, and therefore, suspended from donations. Continuing the complete remission, it was decided to attempt the collection of autologous peripheral stem cells free from disease, for a possible second approach with HDT; unfortunately without success.

The potential donor (the female cousin) was again summoned at the end of November 2020. Meanwhile, crizotinib therapy was continued and the AIFA’s approval requested for its continuation. Pending the authorization of the AIFA, we plan to continue crizotinib therapy until allo-SCT can be performed, provided that the response to this drug continues to be good. Although we do not have any data on allo-SCT after this targeted therapy with ALK-TKI, we also do not have data that advise against this treatment before allo-SCT. Therefore, it seems to be the best strategy that can be used right now.

## 3. Statement of Ethics

All procedures performed were in accordance with the international guidelines, with the Helsinki Declaration of 1975, revised in 1983, and the roles of the Italian laws of privacy. Crizotinib treatment was approved by the AIFA (Agenzia Italiana del Farmaco) on 7 February 2020.

The patient signed an anonymous informed consent form for the use of his data for anonymous clinical investigations and scientific publications.

## 4. Discussion

The time we live in is particularly dramatic worldwide, due to the SARS-CoV-2 pandemic. Hospital assistance is in trouble everywhere and everyday hospital structures previously used for the various sectors of medicine have been adapted into centers for the reception of patients affected by SARS-CoV-2. There are severe limitations to outpatient and day hospital activities and, in many local situations, the access to hospitals for neoplastic or otherwise immunosuppressed patients has been prevented or severely slowed for fear of contagion. Equally slowed down has been the process for accessing organ transplantation, and transplant operations have become infrequent and limited to situations of emergency [29,30].

In Italy, these conditions have been aggravated by a prolonged organizational inefficiency of the healthcare governing institutions, which have caused a substantial reduction in the number of doctors, nurses, and all other categories of health workers and a marked reduction in the number of beds—conditions known for years but dramatically enhanced by the SARS-CoV-2 pandemic. In these conditions, it is complex to put into practice the indications of international guidelines and to avoid that the infection by SARS-CoV-2 increases the risk of mortality in neoplastic patients.

After two relapses, our patient achieved complete remission of the disease with brentuximab used as a third-line therapy. According to international guidelines, patients with ALCL ALK +, once a CR at a rescue therapy has been obtained, must undergo allo-SCT, which could have a high chance of success [31,32,33,34,35]. The young age of the patient and her refractory to the first- and second-line treatments prompted us to proceed toward allo-SCT, so far without success, due to the unsuitability of the contacted donors [36,37,38,39]. Although in fair condition after 16 cycles of Brentuximab, fearing that even this third-line treatment would fail while waiting for a suitable donor, we decided to use crizotinib [40,41,42,43], a drug that, in Italy, is not registered for this type of lymphoma, but with a specific action against ALK + and effective in published anecdotal cases.

Already in 2013, Gambacorti Passerini described encouraging results obtained with crizotinib in patients affected by ALCL ALK + and who relapsed in multiple therapeutic lines. Since then, however, the trend was to continue using the drugs used before for other neoplasms and only recently have a few cases of ALCL ALK + treated with crizotinib been described [44,45].

## 5. Conclusions

In conclusion, in regard to our patient, the fear of a failure of the third-line therapy (brentuximab), the difficulties encountered in identifying a compatible donor, and the poor stability of the hospital network during this pandemic led us to treat our patient with crizotinib 500 mg/day. Excellent results have been obtained, a persistent CR, and an excellent clinical condition both witnessed by a perfectly normal [18f] FDG-PET-CT, and an excellent psycho-emotional condition that sustains her while waiting for allo-SCT.

Our judgment on the use of crizotinib therapy in ALCL ALK + treated is excellent and we believe it should be included in guidelines as a bridge to allo-SCT.

## Figures and Tables

**Figure 1 healthcare-09-00135-f001:**
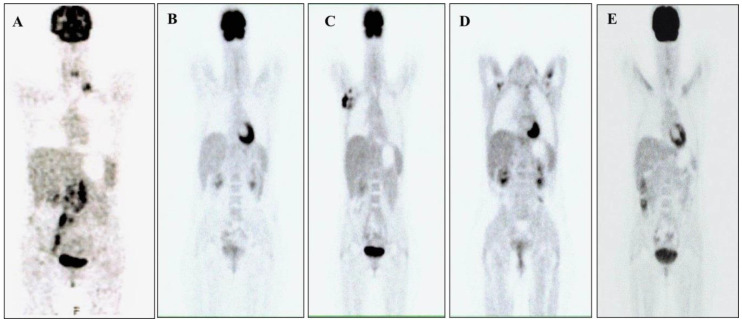
(**A**) 18 F-fluorodeoxyglucose- Positron emission tomography–computed tomography ([18f] FDG-PET-CT) scans performed in October 2016; the examination showed pathologies with a high glucose metabolism in correspondence with multiple lymph node packets in the left supraclavicular region, retroclaveare left and right, left retrocrural, lomboaortica, intercavale, paracavale, the iliac bifurcation, and along the path of the internal inguinal vessels and right exteriors. (**B**) [18f] FDG-PET-CT scans performed in January 2018; the examination did not show pathologies with a high glucose metabolism. (**C**) [18f] FDG-PET-CT scans performed in July 2019; the examination showed pathologies with a high glucose metabolism in correspondence with numerous lymphadenomegalies in the right axillary region and of the lymph nodes in the right lung perilary area. (**D**) [18f] FDG-PET-CT scans performed in June 2020; the examination did not show pathologies with a high glucose metabolism. (**E**) [18f] FDG-PET-CT scans performed in September 2020; the examination did not show pathologies with a high glucose metabolism.

**Table 1 healthcare-09-00135-t001:** Treatment timeline.

Period	Therapy
From October 2016 to February 2017	CEOP: 6 cycles
From April 2017 to August 2017	MTX-ARAC-THIOthepa: 4 cycles plus 6 rachicentesis with infusion of MTX, ARAC and steroids
From October 2017 to November 2017	ASCT after HDT with Thio-Bu
From July 2019 to June 2020	Brentuximab: 16 cycles
From July 2020 to present	Crizotinib: 250 mg, twice daily

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
