# Peer review of "How Fear of COVID-19 Can Affect Treatment Choices for Anaplastic Large Cell Lymphomas ALK+ Therapy: A Case Report"

_healthcare, 2021, doi:10.3390/healthcare9020135_

Round 1

Reviewer 1 Report

Sica A. & colleagues have provided an interesting rapport about off-label use of ALK-TKI, Crizotinib, for a patient with ALK positive Anaplastic Large Cell Lymphoma (ALCL). The ALK-rearrangement was constituted by the most common fusion partner, Nucleophosmin 1 gene, made a fusion with ALK gene to NPM1-ALK rearrangement, which is occurring in about 80% of ALK positive cases of ALCL.
Since having refractory disease and under the COVID-19 pandemic affecting the possibility of the further standard treatment, the targeted therapy with 1. generation ALK-TKI, Crizotinib, was decided to the patient, and maintaining of complete response was observed. The authors have considered the treatment with Crizotinib as a bridge the patient could be staying safe until the definitive treatment will be possible to achieve.  Crizotinib was therefore used as a 4th line of treatment and started under complete response to the previous 3rd line with Brentuximab.
This patient story tells us that life sometime forces us to make difficult choices, but also to revise approach to the data we already have. There is a clear ethical aspect: we do not have enough data which can help to make a good decision, but we have a patient we must address the problem best we can. Furthermore, it opens a discussion about interpretation and significance of the presence of different oncodrivers in different diagnoses and ages. In the current state of knowledge, it seems that the presence of oncodriver prevail the phenotype/diagnosis. Current observation shows that Crizotinib can be e.g., effective in pediatric patients with ALK positive ALCL (1) or infantile hemispheric glioma with ALK -rearrangement can successfully treated with Crizotinib (2). Other case reports in adults with ALK -rearrangement other than described in NSCLC showed also response to ALK-TKI as it was reported for ganglioneuroblastoma (3).
Regarding ALK positive ALCL, there are, beyond NPM1, many other fusion partners like: TPM3, TFG, ATIC, CLTC, TPM4, MSN, ALO17, MYH9, TRAF1 and TPM3 described (4,5). As we have learned from non-small cell lung cancer (NSCLC), the fusion partner can have influence on response to different ALK-TKI, so we still need to explore how it works in other diagnoses with ALK-rearrangement (6). There are currently ongoing several clinical trials e.g., Lorlatinib in refractory ALK positive ALCL (NCT03505554) and in refractory neuroblastoma (NCT03107988) or Brigatinib in ALK positive ALCL (NCT03719898), and they probably can bring more knowledge how ALK-TKI work in other diagnoses like ALCL or other solid tumors with ALK-rearrangement.
This patient case contributes to our knowledge of the effect of ALK-TKI in other diagnoses than ALK positive NSCLC and illustrates how, in the new realities, we can learn new approaches and adopt them to the best interest of patient.

References:
1. Mossé YP, Voss SD, Lim MS, Rolland D, Minard CG, Fox E, Adamson P, Wilner K, Blaney SM, Weigel BJ. Targeting ALK With Crizotinib in Pediatric Anaplastic Large Cell Lymphoma and Inflammatory Myofibroblastic Tumor: A Children's Oncology Group Study. J Clin Oncol. 2017 Oct 1;35(28):3215-3221. doi: 10.1200/JCO.2017.73.4830. Epub 2017 Aug 8. PMID: 28787259; PMCID: PMC5617123.

  1. Guerreiro Stucklin, A.S., Ryall, S., Fukuoka, K. et al. Alterations in ALK/ROS1/NTRK/MET drive a group of infantile hemispheric gliomas. Nat Commun 10, 4343 (2019). https://doi.org/10.1038/s41467-019-12187-5
  2. Risum S, Knigge U, Langer SW. Hitherto unseen survival in an ALK-positive-patient with advanced stage adult ganglioneuroblastoma treated with personalized medicine. Clin Case Rep. 2017 Nov 7;5(12):2085-2087. doi: 10.1002/ccr3.1262
  3. Tsuyama N et al.  Anaplastic large cell lymphoma: pathology, genetics, and clinical aspects.  Journal of clinical and experimental hematopathology, Vol. 57 No.3, 120-142, 2017, and
  1. B. Hallberg* & R. H. Palmer. The role of the ALK receptor in cancer biology. Annals of Oncology 27 (Supplement 3): iii4–iii15, 2016, doi:10.1093/annonc/mdw301).
  2. Childress MA, Himmelberg SM, Chen H, Deng W, Davies MA, Lovly CM. ALK Fusion Partners Impact Response to ALK Inhibition: Differential Effects on Sensitivity, Cellular Phenotypes, and Biochemical Properties. Mol Cancer Res. 2018 Nov;16(11):1724-1736. doi: 10.1158/1541-7786.MCR-18-0171. 

I have a couple of questions:

  1. What was the variant (pattern) of patient´s ALCL?
  2. How was the NPM1-ALK fusion identified (IHC, FISH, NGS (which panel)? Has it been any discrepancy between the methods used?
  3. Please explain more about the condition in the line 79-80 that: A voluminous lymphadenopathy appeared in the right armpit at the end of May 2019, greatly reduced with paracetamol. Do you interpret it as an early sign of progression, where NSAID can provide a transient improving?
  4. How long was the gap between the last dose of Brentuximab and start on Crizotinib?
  5. Please explain for the readers what was the reason to stop Brentuximab under complete response?
  6. I suggest the treatment timeline would be easier to follow by making a chart/diagram
  7. What are the future therapeutic plans for the patient? Are you going to treat the patient with Crizotinib until progression and then planning allo-SCT? Or you still regard treatment with Crizotinib as a bridge to allo-SCT, since we do not have any data how allo-SCT will work after targeted therapy with ALK-TKI?

Minor corrections:

- line 40: double space between Cell and Lymphomas

- line 25: rephase the sentence for improving the clarity of wording: chromosomal rearrangement and resulting nucleophosmin (NPM1) -ALK fusion. It is chromosomal rearrangement of ALK gene with Nucleophosmin1 gene (NPM1) resulting in NPM1-ALK fusion.

- line 68: Centre is in capital letter. If it is not the name of your hospital, then lower case letter is enough

-line 70: please add “classification”. Otherwise, Ann Arbor is the city in the U.S. state of Michigan.

- line 71: Figures should be inserted close to their first citation; so please move to line 71

- line 80: expand the acronym VES

- line 90: please rephrase the sentence: One a CR was achieved (..). Meaning will be: Since (or as) CR was achieved, 

- line 145: please rephrase the sentence: the trend has been to direct the use of this drug.
The meaning is that the trend was directed towards the use of the drug.

- line 150: double space between mg / day.

Reviewer 2 Report

The author is to make the introduction a little simple by providing the details about their reasoning for selecting Crizonitib drug for treatment as this drug is not registered in Italy.

It is also not clear what ALK inhibitors should be used for treating ALCL, but AIFA allows only for non-Small Cell  Lung Cancer. It will be good if they provide reasoning on why AIFA registered these drugs for lung cancer. Also, they should give the name of such drugs.

Reviewer 3 Report

Sica and coworkers presented a case report of a woman affected by ALK+ Anaplastic Large Cell Lymphomas waiting for bone marrow transplantation. In particular, the authors have described the therapeutic plan of the woman as well as her medical history, highlighting the use of Crizotinib as an off-label treatment of ALCL. In particular, the authors suggest that crizotinib should be approved also for the treatment of ALK+ ALCL. It is not clear what is the link between the COVID-19 pandemic and the case report here described. Below are reported some minor/major comments that the authors have to address before publication:
1) The main critical issue is related to the link between the COVID-19 pandemic and the case report described by the authors. How COVID-19 pandemic altered the therapeutic option of the patient? Has the pandemic slowed down the search for donors or prevented the patient from accessing hospital services? Please better argue these aspects as in the Title it is clearly stated “How fear of COVID-19 can affect treatment choices….”;
2) Please better rewrite the following sentence: “One a CR was achieved after 6 months of this third line of treatment, the donor selection for allo-SCT began.”;
3) Please provide references for the following paragraph: “The time we live in is particularly dramatic worldwide due to the SARS-CoV-2 pandemic. Hospital assistance is in trouble everywhere and everyday hospital structures previously used for the various sectors of medicine are adapted into centers for the reception of patients affected by SARS-CoV-2. There are severe limitations to outpatient and day hospital activities and in many local situations the access to hospitals for neoplastic or otherwise immunosuppressed patients is prevented or severely slowed for fear of contagion. Equally slowed down is the process for accessing organ transplantation and transplant operations have become infrequent and limited to situations of emergency. In Italy, these conditions are aggravated by a prolonged organizational inefficiency of the healthcare governing institutions, which have caused a substantial reduction in the number of doctors, nurses and all other categories of health workers and a marked reduction in the number of beds, conditions known from years, but dramatically enhanced by the SARS-CoV-2 pandemic. In these conditions it is complex to put into practice the indications of international guidelines and to avoid that the infection by SARS-CoV-2 increases the risk of mortality in neoplastic patients.” For this purpose, see:
– 10.3390/cancers12082237
– 10.1097/COC.0000000000000712
– 10.3892/ijo.2020.5159
4) In the Discussion section, please provide more references about crizotinib and its application. For this purpose, see:
- 10.12659/AJCR.903528
- 10.3389/fphar.2018.01300
- 10.2147/DDDT.S91988
- 10.1186/1471-2407-14-683
5) The manuscript should be revised by an English native speaker.
